# Adaptive variational quantum minimally entangled typical thermal states for finite temperature simulations

João C. Getelina[1], Niladri Gomes[1†], Thomas Iadecola[1,2],
Peter P. Orth[1,2,3] and Yong-Xin Yao[1,2⋆]

**1** Ames National Laboratory, U.S. Department of Energy, Ames, Iowa 50011, USA
**2** Department of Physics and Astronomy, Iowa State University, Ames, Iowa 50011, USA
**3** Department of Physics, Saarland University, 66123 Saarbrücken, Germany

⋆ ykent@iastate.edu

## Abstract

Scalable quantum algorithms for the simulation of quantum many-body systems in thermal equilibrium are important for predicting properties of quantum matter at finite temperatures. Here we describe and benchmark a quantum computing version of the minimally entangled typical thermal states (METTS) algorithm for which we adopt an adaptive variational approach to perform the required quantum imaginary time evolution. The algorithm, which we name AVQMETTS, dynamically generates compact and problem-specific quantum circuits, which are suitable for noisy intermediate-scale quantum (NISQ) hardware. We benchmark AVQMETTS on statevector simulators and perform thermal energy calculations of integrable and nonintegrable quantum spin models in one and two dimensions and demonstrate an approximately linear system-size scaling of the circuit complexity. We further map out the finite-temperature phase transition line of the two-dimensional transverse field Ising model. Finally, we study the impact of noise on AVQMETTS calculations using a phenomenological noise model.

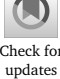

## Contents

---

† Present address: Lawrence Berkeley National Laboratory, Berkeley, CA 94720, USA

# 1  Introduction

Theoretically deriving finite-temperature thermodynamic properties of interacting quantum many-body systems is notoriously difficult, but important to be able to compare to experimental results. Determining the behavior near and at thermal phase transitions, including a prediction of transition temperatures, is one of the central challenges of condensed matter physics. There are several computational techniques that address this question. Exact diagonalization (ED), for example, is generally applicable and can be used at arbitrary temperature, if the full energy spectrum is computed [1]. However, ED is limited to small system sizes, especially if many excited states contribute at finite temperature. An alternative approach is based on thermal pure quantum (TPQ) states [2,3], which significantly reduces the computational load compared to ED. It is still constrained in the system size that can be simulated as it requires the preparation and evolution of a random quantum state in Hilbert space, which grows exponentially with system size. Techniques that work well at high temperatures are Quantum Monte Carlo approaches [4–6], high-temperature and numerical linked-cluster expansion methods [7–9], as well as the pseudo-Majorana functional renormalization group method [10,11]. They become less reliable at lower temperatures, where spatial correlations build up. Finally, the numerical renormalization group [12,13] is well suited for the description of quantum impurity models at low temperatures, which can be combined with embedding methods such as the dynamical mean-field theory to simulate lattice models [14–17].

One of most powerful numerical techniques for low-dimensional systems is based on approximating the many-body wavefunction as a matrix product state (MPS). This underlies the density matrix renormalization group (DMRG) method [18] and generally tensor network approaches [19]. These methods can also be applied to simulate systems at finite temperature, for example, by using a ancilla purification method [20,21]. This approach introduces a reference system with the same dimension as the physical system of interest, and prepares a state where the physical and reference systems are maximally entangled. The physical system is then evolved in imaginary time until a given $\beta = 1/T$, where it reaches a purified Gibbs state at temperature $T$, which can be used to calculate thermodynamic quantities. This approach is conceptually appealing, but requires a significantly larger bond dimension $\chi'$ to represent the joint state of system and ancillae compared to the bond dimension $\chi$ required for the pure system simulation. In the limit $\beta \to \infty$ the difference approaches $\chi' = \chi^2$ [22]. An alternative technique that circumvents this challenge relies on sampling minimally entangled typical thermal states (METTS) [22–24], which are states obtained by imaginary-time evolution of product states. Since this method only evolves the physical system, the computational complexity to generate a METTS is comparable to that of a ground state DMRG calculation. On the other hand, it requires sampling over potentially many METTSs depending on the simulation temperature. The METTS technique is therefore most suited for simulating the low temperature regime [25].

To address some of the limiting issues of classical algorithms, several quantum algorithms for finite-temperature simulations of quantum many-body systems have been proposed [26–39]. Some algorithms focus on preparing purified thermal states using variational approaches [32,35,37]. This requires the classical optimization of a cost function and provides a quantum generalization of the ancilla purification method discussed above [20,21]. A quantum analog of the METTS algorithm (QMETTS) has also been proposed [34,38] and utilizes a quantum imaginary time evolution (QITE) algorithm to evolve the initial product states [34]. This method offers the potential advantage of requiring exponentially less space and time resources to store and evolve quantum states of 2D and 3D systems compared to the METTS algorithm using MPSs. We note that, while it might seem straightforward and appealing to leverage the statistical approach based on TPQ states in a quantum algorithm, the necessary initial step of random state preparation is known to be exponentially hard on quantum computers [40].

The quantum resource requirements for QMETTS are determined by QITE for which the quantum circuit depth scales exponentially with the system correlation length $\xi$ and it grows linearly with imaginary time $\beta$. Several techniques have been developed to reduce the circuit complexity for practical QITE calculations [38,41,42] to make it more suited for noisy intermediate-scale quantum (NISQ) hardware. However, the execution of QITE on current quantum hardware remains limited to small system sizes so far. Another related approach is variational quantum imaginary time evolution based on MacLachlan's principle, which corresponds to following the quantum natural gradient descent path [43–47]. This approach expresses the state by a parametrized quantum circuit whose variational parameters are updated according to a classical equation of motion, whose coefficients are obtained on a quantum computer. It was shown that an adaptive construction of the variational ansatz yields compact and problem specific parametrized circuits that enable high fidelity quantum state propagation in imaginary time (AVQITE) [47]. The AVQITE algorithm adaptively expands the variational circuit ansatz with unitaries drawn from a predefined operator pool in order to maintain a high state fidelity at each timestep along the imaginary time path.

In this work, we use AVQITE to perform imaginary time evolution in an adaptive variational quantum minimally entangled typical thermal states (AVQMETTS) algorithm. The AVQMETTS approach leverages the shallow circuits produced by AVQITE and is thus promising for application on NISQ hardware. As a first step, we here perform benchmark calculations on statevector simulators and show that AVQMETTS calculations can yield accurate thermal expectation values, for example, the thermal energy. We investigate thermal properties of the transverse-field Ising model (TFIM) and the mixed-field Ising model (MFIM) with up to 20 sites in one-dimensional (1D) and two-dimensional (2D) lattices. For 1D models, we observe linear system-size scaling of the circuit complexity as characterized by the number of CNOT gates. This agrees with the scaling observed in METTS calculations using MPSs for gapped 1D systems, if the MPS is written as a quantum circuit [48]. For 2D models, we observe that the circuit complexity scales approximately linear for the MFIM and superlinear for the TFIM. This implies a potential advantage of AVQMETTS calculations for 2D systems compared to METTS calculations, where the bond dimension of MPSs generally scales exponentially with system size [18]. Furthermore, we use AVQMETTS to determine several points on the finite temperature phase boundary between ferromagnetic and paramagnetic phases in the 2D TFIM. Finally, we implement noisy AVQMETTS simulations based on a phenomenological noise model as a preliminary investigation of noise effects.

The remainder of the article is organized as follows: in Sec. 2, we describe the AVQMETTS algorithm, and in Sec. 3 we introduce the TFIM and MFIM together with the operator pool used for AVQMETTS. We analyze the performance of the algorithm at a few representative thermal steps in Sec. 4, before presenting results for the spin models in 1D in Sec. 5 and 2D in Sec. 6. In Sec. 7, we present results of the Binder cumulant and determine the critical temperature in

the TFIM. The noisy simulation results are discussed in Sec. 8. Conclusions and outlook are given in Sec. 9.

## 2  AVQMETTS algorithm

We are interested in calculating the thermal expectation value of an observable $\hat{\mathcal{O}}$ in an $N$-qubit quantum system at inverse temperature $\beta = 1/T$. Assuming that the system is governed by a Hamiltonian $\hat{\mathcal{H}}$ the thermal expectation value can be obtained from

$$\langle \hat{\mathcal{O}} \rangle_\beta = \frac{1}{\mathcal{Z}} Tr e^{-\beta \hat{\mathcal{H}}} \hat{\mathcal{O}} = \frac{1}{\mathcal{Z}} \sum_{i=1}^{2^N} P_i \mathcal{O}_i(\beta). \tag{1}$$

Here, we have defined the expectation value $\mathcal{O}_i(\beta) \equiv \langle \phi_i(\beta) | \hat{\mathcal{O}} | \phi_i(\beta) \rangle$ in a METTS

$$|\phi_i(\beta)\rangle = P_i^{-1/2} e^{-\beta \hat{\mathcal{H}}/2} |i\rangle , \tag{2}$$

which is obtained by imaginary time evolution, starting from a classical product state (CPS) $|i\rangle$. The summation over $i$ in Eq. (1) runs over a complete CPS basis. The probability distribution of METTSs is proportional to $P_i = \langle i | e^{-\beta \hat{\mathcal{H}}} | i \rangle$ and is normalized by the canonical partition function $\mathcal{Z} = \sum_i P_i$. In the METTS algorithm, the thermal average (1) can be efficiently evaluated using a METTS ensemble of size $S$ as

$$\langle \hat{\mathcal{O}} \rangle_\beta \approx \frac{1}{S} \sum_{i=1}^{S} \mathcal{O}_i(\beta). \tag{3}$$

As described in detail below, the METTS ensemble is generated by independent Markovian random walks, which samples METTS with the desired distribution $P_i/\mathcal{Z}$ as a fixed point in this process [22, 23].

In the AVQMETTS approach, one uses AVQITE for state propagation along imaginary time $\tau$ from a CPS $|i\rangle$ at the initial time $\tau = 0$ to the corresponding METTS $|\phi_i(\beta)\rangle$ at final time $\tau = \beta/2$. It was shown previously that AVQITE produces compact variational quantum states that allow for imaginary time evolution with high fidelity [47]. This is achieved by representing the time-dependent quantum state with an adaptive variational ansatz in a pseudo-Trotter form

$$|\phi_i[\boldsymbol{\theta}(\tau)]\rangle = \prod_{\mu=1}^{N_\theta} e^{-i\theta_\mu(\tau)\hat{A}_\mu} |i\rangle . \tag{4}$$

Here, $\hat{A}_\mu \in \{I, X, Y, Z\}^{\otimes N}$ is a Pauli string defined as a direct product of Pauli operators for an $N$-qubit system. The time evolution of the quantum state is encoded in the variational parameters $\boldsymbol{\theta}(\tau)$, which propagate according to the equations of motion determined by the McLachlan variational principle [43, 44, 49]

$$\frac{d\boldsymbol{\theta}}{d\tau} = \mathbf{M}^{-1}\mathbf{V}. \tag{5}$$

Here we define the quantum Fisher information matrix

$$M_{\mu\nu} = 2\Re \left[ \frac{\partial \langle \phi[\boldsymbol{\theta}]|}{\partial \theta_\mu} \frac{\partial |\phi[\boldsymbol{\theta}]\rangle}{\partial \theta_\nu} - \frac{\partial \langle \phi[\boldsymbol{\theta}]|}{\partial \theta_\mu} |\phi[\boldsymbol{\theta}]\rangle \langle \phi[\boldsymbol{\theta}]| \frac{\partial |\phi[\boldsymbol{\theta}]\rangle}{\partial \theta_\nu} \right],$$

and the energy gradient $V_\mu = 2\Re \left[ -\frac{\partial \langle \phi[\boldsymbol{\theta}]|}{\partial \theta_\mu} \hat{\mathcal{H}} |\phi[\boldsymbol{\theta}]\rangle \right]$, which can be measured on a quantum computer. A detailed discussions on quantum circuit implementations is given in Refs. [47, 50].

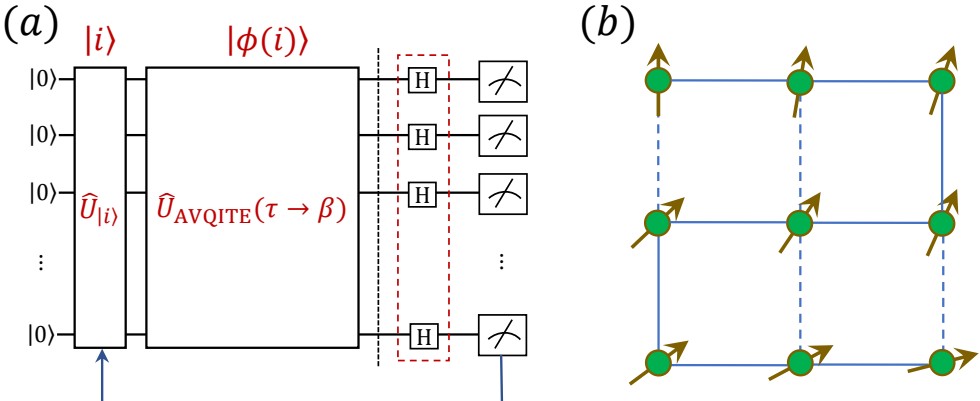

Figure 1: **AVQMETTS approach and model setup.** (a) Schematic flowchart of the AVQMETTS algorithm, which includes application of quantum circuits $\hat{U}_{|i\rangle}$ for initial CPS $|i\rangle$ preparation in the $X$- or $Z$-basis, followed by application of AVQITE circuits for imaginary time evolution to a desired inverse temperature $\beta$. The final Hadamard gates realize a measurement basis rotation that is applied in every other thermal step, and $Z$ basis measurements are performed at the end of each thermal step. The resulting bitstring is then used as the initial CPS during the next thermal step. (b) Layout of the 1D (9-site) or 2D (3×3) lattice of the MFIM. The 2D model is obtained by considering the dashed couplings with the same coupling as the solid ones. Periodic boundary conditions are used for both models, and applied in both directions for the 2D lattice.

We note that we here implement the algorithm fully classically to benchmark its performance in the absence of noise. The equations of motion (5) are integrated using the Euler method with a time step $\delta\tau$ that controls the simulation duration and accuracy. We note that the resulting AVQITE circuit is problem-specific and tied to the initial state $|i\rangle$.

In the AVQITE approach, the set of $N_{\boldsymbol{\theta}}$ generators $\{\hat{A}_\mu\}$ in the variational ansatz in Eq. (4) is constructed automatically and can be gradually expanded along the dynamical path by appending optimal Pauli strings from a predefined operator pool $\mathscr{P}$ of Pauli strings. The ansatz expansion is performed such that the McLachlan distance [43, 47] remains below a desired threshold $L_{\text{cut}}$. The McLachlan distance is a measure of the difference between the variational state evolution and the exact one during a single time step.

The flowchart of AVQMETTS is illustrated in Fig. 1(a). An initial thermal step of the algorithm starts with a random CPS $|i\rangle$ in the computational $Z$-basis prepared with a quantum circuit. The AVQITE algorithm is then employed to represent the METTS $|\phi_i[\boldsymbol{\theta}(\beta)]\rangle$ in terms of a parametrized circuit, which allows for a measurement of the expectation value $\mathcal{O}_i[\boldsymbol{\theta}(\beta)] \equiv \langle\phi_i[\boldsymbol{\theta}(\beta)]|\hat{\mathcal{O}}|\phi_i[\boldsymbol{\theta}(\beta)]\rangle$. The next thermal step follows after collapsing the METTS to a new CPS $|i'\rangle$ with probability $|\langle\phi_i[\boldsymbol{\theta}(\beta)]|i'\rangle|^2$ through a quantum measurement. The result of time-propagation followed by state collapse is a Markovian random walk between METTSs corresponding to different initial CPSs. As the AVQITE circuit is associated with the initial state, each distinct CPS requires a unique AVQITE calculation. However, reusing AVQITE circuits is also feasible and can help minimize the quantum resource demand for AVQMETTS calculations. This is due to the fact that a CPS obtained from state collapse after a thermal step may be identical to one that was sampled in a previous step due to the inherent structure of the distribution of METTS. In the numerical simulations reported below, we observe $10 - 60\%$ of CPSs are sampled for more then once, depending on the system size and temperature. Following earlier results, we apply alternating $X$- and $Z$-basis measurements in consecutive thermal

steps to effectively reduce the autocorrelation time of the walk [23]. In practice, a METTS ensemble of size $S = S_{\rm w}S_0$ is obtained using $S_{\rm w}$ independent Markovian random walks, which can be executed in parallel, each of which generates $S_0$ METTSs. We discard the initial thermal steps, typically the first ten, in each random walk to erase memory effects. The METTS ensemble average gives an efficient estimation of the thermal expectation value

$$\langle \hat{\mathcal{O}} \rangle_\beta \approx \langle \hat{\mathcal{O}} \rangle_{\boldsymbol{\theta}(\beta)} \equiv \frac{1}{S} \sum_i \mathcal{O}_i[\boldsymbol{\theta}(\beta)], \tag{6}$$

which is the quantity we are interested in.

## 3 Spin models

To benchmark the accuracy and scalability of the AVQMETTS approach, we consider the non-integrable mixed-field nearest-neighbor Ising model in 1D and on the 2D square lattice with periodic boundary conditions (PBC) as illustrated in Fig. 1(b). The Hamiltonian reads

$$\hat{\mathcal{H}} = -J \sum_{\langle jk \rangle} \hat{Z}_j \hat{Z}_k - \sum_j \left( h_x \hat{X}_j + h_z \hat{Z}_j \right), \tag{7}$$

where $\langle jk \rangle$ denotes summation over nearest neighbors. In the following, we consider the ferromagnetic model ($J > 0$) and measure the energy in units of $J = 1$. The 1D MFIM reduces to the integrable TFIM when the longitudinal field vanishes $h_z = 0$. The competition between ferromagnetic and paramagnetic phases is controlled by the transverse field $h_x$ and the temperature $T$. The 1D TFIM exhibits a quantum critical point at $h_x = 1$ and $T = 0$. While the 1D TFIM is paramagnetic at any finite temperature $T > 0$ and only undergoes thermal crossovers [51,52], the 2D TFIM undergoes a continuous phase transition in the Ising universality class in the $(h_x, T)$ parameter plane. At $h_x = 0$, the model reduces to the classical Ising model on the square lattice with a critical temperature $T_c = 2J/\ln(1 + \sqrt{2})$ given by Onsager's exact solution [53]. The transition temperature $T_c(h_x)$ decreases continuously with increasing $h_x > 0$ and reaches $T_c = 0$ at the quantum critical point $h_x/J = 3.04438$ [54]. In contrast, the ferromagnetic MFIM only exhibits crossovers at zero and finite temperatures due to the explicit breaking of the global $\mathbb{Z}_2$ symmetry $Z_i \to -Z_i$ by the longitudinal field [55].

In the following, we focus on two sets of model parameters, where we apply AVQMETTS to evaluate the thermal energy. First, we consider the 1D and 2D TFIM at finite temperatures $T > 0$ above the quantum critical point, with $(h_x, h_z) = (1, 0)$ for 1D and $(3.05, 0)$ for 2D. Second, we investigate the 1D and 2D MFIM, where we keep the same transverse field $h_x$ as the TFIM, and set $h_z = h_x/2$. We consider system sizes in the range $4 \leq N \leq 20$.

When constructing the adaptive ansatz, we use the following complete operator pool [56, 57]:

$$\mathscr{P} = \{Y_j\}_{j=1}^N \cup \{Y_j Z_k, Z_j Y_k\}_{1 \leq j < k \leq N}. \tag{8}$$

This pool is composed of all one-qubit and two-qubit Pauli strings that contain a single $Y$ Pauli matrix. Since every generator contains an odd number of $Y$ operators, a variational wavefunction that is initialized with real coefficients remains real along the imaginary-time path. In this case, the second term in the quantum Fisher information matrix $M_{\mu\nu}$ vanishes [47].

## 4 Analysis of representative thermal steps

The accuracy of the variational state preparation in AVQMETTS using AVQITE and the associated circuit complexity is analyzed in Fig. 2 for two representative thermal steps in sim-

ulations of a $N = 14$ site MFIM. The model parameters are $J = 1$, $h_x = 1$, $h_z = 0.5$, and the simulated temperatures are in the range $0.2 \leq \beta \leq 4$. In the AVQITE calculations, we use the time step $\delta\tau = 0.02$ and the MacLachlan distance threshold $L_{\text{cut}} = 0.001$. Results are shown for a thermal step with an initial $Z$-basis CPS (green) and an initial $X$-basis CPS (blue), which are obtained by a measurement in the $Z$ and $X$-basis to collapse a METTS at $\beta = 2$. We refer to them as $Z$- and $X$-thermal steps hereafter. Clearly, one observes a characteristic difference between these two thermal steps. In Fig. 2(a), we compare the energy of the exact imaginary-time-evolved state $E_i(\tau) = \langle\phi_i(\tau)|\hat{\mathcal{H}}|\phi_i(\tau)\rangle$ with the variational energy $E_i[\boldsymbol{\theta}(\tau)] = \langle\phi_i[\boldsymbol{\theta}(\tau)]|\hat{\mathcal{H}}|\phi_i[\boldsymbol{\theta}(\tau)]\rangle$. The absolute energy difference is generally smaller than 0.1. The $Z$-thermal step produces a smaller error than in the $X$-thermal step up to a final propagation time of $\tau \approx \beta = 2$. For $\tau \geq 2$, we observe a convergence of the absolute energy difference within about $10^{-4}$ to $10^{-6}$. The inset shows the scale of the energy, from which we derive that the relatively error is smaller than 0.5%. Generally, the relatively larger error in the $X$-thermal step correlates with the higher energy of the initial $X$-basis CPS at $\tau = 0$, as plotted in the inset.

Fig. 2(b) shows that the associated state infidelity is generally smaller than $10^{-3}$. At $\tau = \beta = 2$, the infidelity reaches $2 \times 10^{-5}$ at $Z$-thermal step and $6 \times 10^{-5}$ at $X$-thermal step. For $\tau \geq 2$, the infidelity shows a convergence to within about $2 \times 10^{-5}$ to $10^{-6}$. The overall improvement of the variational state energy error and infidelity with increasing $\beta$ implies that the effective reduction of Hilbert space dimension due to lowering temperature dominates over the error accumulation in discretized state propagation due to the finite timestep $\delta\tau$ in AVQITE.

In Fig. 2(c) we plot the $\tau$-dependence of the number of variational parameters $N_{\boldsymbol{\theta}}$ in the pseudo-Trotter state. Each parameter is associated with a generator selected from the operator pool $\mathcal{P}$ defined in Eq. (8). Generally, the number of parameters $N_{\boldsymbol{\theta}}$ grows with $\tau$ as more variational degrees of freedom are required to cover the entire dynamical path. The difference of the state energy that we observe during $Z$- and $X$-thermal steps is also manifested in a disparity in the number of variational parameters: for $\tau > 1$, $N_{\boldsymbol{\theta}} \approx 100$ for the $X$-thermal step, which is about 2.5 times larger than for the $Z$-thermal step. Finally, to further characterize the circuit complexity for NISQ applications, we plot in Fig. 2(d) the number of CNOT gates $N_{\text{cx}}$ required to prepare the variational pseudo-Trotter state as a function of $\tau$. The behavior of $N_{\text{cx}}$ mirrors that of $N_{\boldsymbol{\theta}}$. At large $\tau > 1$, $N_{\text{cx}} \approx 150$ for the $X$-thermal step, which is about three times larger than $N_{\text{cx}}$ for the $Z$-thermal step. In this analysis, we assume that the quantum device has all-to-all qubit connectivity, which allows us to simplify the calculation by considering a two-qubit rotation gate as requiring $N_{\text{cx}} = 2$ CNOTs. Since $N_{\text{cx}}/N_{\boldsymbol{\theta}} \approx 1.5$ when $\tau \geq 2$, we observe that about half of the generators in the adaptive ansätze are two-qubit Pauli strings.

## 5 AVQMETTS results for 1D spin models

In this section, we apply AVQMETTS to the 1D TFIM and MFIM in order to perform a systematic study of the computational accuracy and scalability of the algorithm. The calculation uses $S_{\text{w}}$ independent Markovian random walks that each undergo a number of thermal steps $S_0$ after discarding the initial 10 steps. This generates a METTS ensemble of size $S = S_{\text{w}}S_0$ that we use to estimate thermal averages of observables. Since the statevector simulation time of one thermal step increases with system size, in the following calculations we typically reduce $S_0$ from 128 to 2 for systems with increasing size and correspondingly increase $S_{\text{w}}$ for efficiency. In the following, we focus on the thermal energy $\langle\hat{\mathcal{H}}\rangle_{\beta}$. Fig. 3(a) shows estimated thermal energy $\langle\hat{\mathcal{H}}\rangle_{\boldsymbol{\theta}(\beta)}$ of the $N = 14$ TFIM at $\beta = 2$ (blue) and $\beta = 4$ (green) as a function of thermal step number. The error bars of $\langle\hat{\mathcal{H}}\rangle_{\boldsymbol{\theta}(\beta)}$ indicate the standard error, which is defined as

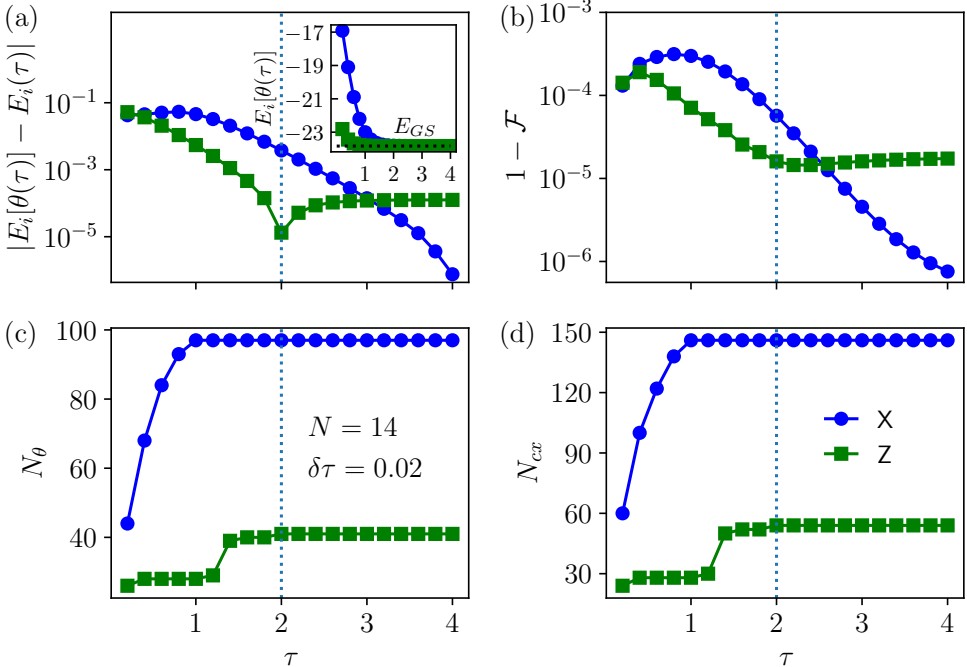

Figure 2: **Accuracy and circuit complexity of two representative thermal steps in AVQMETTS calculations for a 14-site MFIM.** (a) Energy difference between exact and variational state evolution $|E_i[\boldsymbol{\theta}(\tau)] - E_i(\tau)|$ as a function of imaginary time $\tau$ for two representative METTSs obtained for initial $Z$-basis CPS (green) and initial $X$-basis CPS (blue). The CPSs are generated by state collapse from a METTS at $\beta = 2$ as indicated by a vertical dotted line. Inset shows variational energy converging to the ground state energy for large $\tau$. (b) Infidelity $1 - \mathcal{F} = 1 - |\langle \phi(\tau)|\phi[\boldsymbol{\theta}(\tau)]\rangle|^2$ between exact and variational state for the two METTSs versus $\tau$. (c) Number of variational parameters $N_{\boldsymbol{\theta}}$ versus $\tau$. (d) Number of CNOT gates $N_{\mathrm{cx}}$ versus $\tau$. $N_{\mathrm{cx}}$ is calculated assuming all-to-all qubit connectivity as in trapped-ion quantum processors, where each two-qubit rotation gate contributes two CNOTs. Model parameters used are $J = 1$, $h_x = 1$, $h_z = 0.5$, and AVQITE parameters are $\delta \tau = 0.02$ and $L_{\mathrm{cut}} = 10^{-3}$.

$\frac{1}{S} \sqrt{\sum_{i=1}^{S} \left( \mathcal{H}_i[\boldsymbol{\theta}(\beta)] - \langle \hat{\mathcal{H}} \rangle_{\boldsymbol{\theta}(\beta)} \right)^2}$. The model is simulated above the quantum critical point at $h_x = 1$. The dashed line indicates the exact diagonalization (ED) result $\langle \hat{\mathcal{H}} \rangle_{\beta}$. The relatively large variance of $\langle \hat{\mathcal{H}} \rangle_{\boldsymbol{\theta}(\beta)}$ in the first three thermal steps is a manifestation of autocorrelations in the Markov chain, after which $\langle \hat{\mathcal{H}} \rangle_{\boldsymbol{\theta}(\beta)}$ starts to converge. With a fixed ensemble size $S = 288$ in the simulations, the fluctuations of $\langle \hat{\mathcal{H}} \rangle_{\boldsymbol{\theta}(\beta)}$ at $\beta = 4$ are much smaller than those at $\beta = 2$, implying that for a comparable accuracy a smaller sample size is required at lower temperatures due to the reduced number of states that are accessible at lower temperatures.

In contrast, as shown in Fig. 3(b), the AVQMETTS estimation of the thermal energy $\langle \hat{\mathcal{H}} \rangle_{\boldsymbol{\theta}(\beta)}$ converges more rapidly for the 1D MFIM. The estimator also exhibits a smaller error of the mean [note that the vertical scale in this plot is similar to that of Fig. 3(a)]. At $\beta = 2$, no sizable fluctuations in the MFIM simulations are observed after the first thermal step. This can be understood from the energy level diagrams in the inset of (b). A large gap $\sim 4J$ between the ground and first excited states exists for MFIM, while many states are present in this energy window for TFIM. At temperatures $T = 1/\beta$ well below this scale (e.g. $\beta = 2$), the accessible portion of Hilbert space is dominated by the ground state for the MFIM, making the

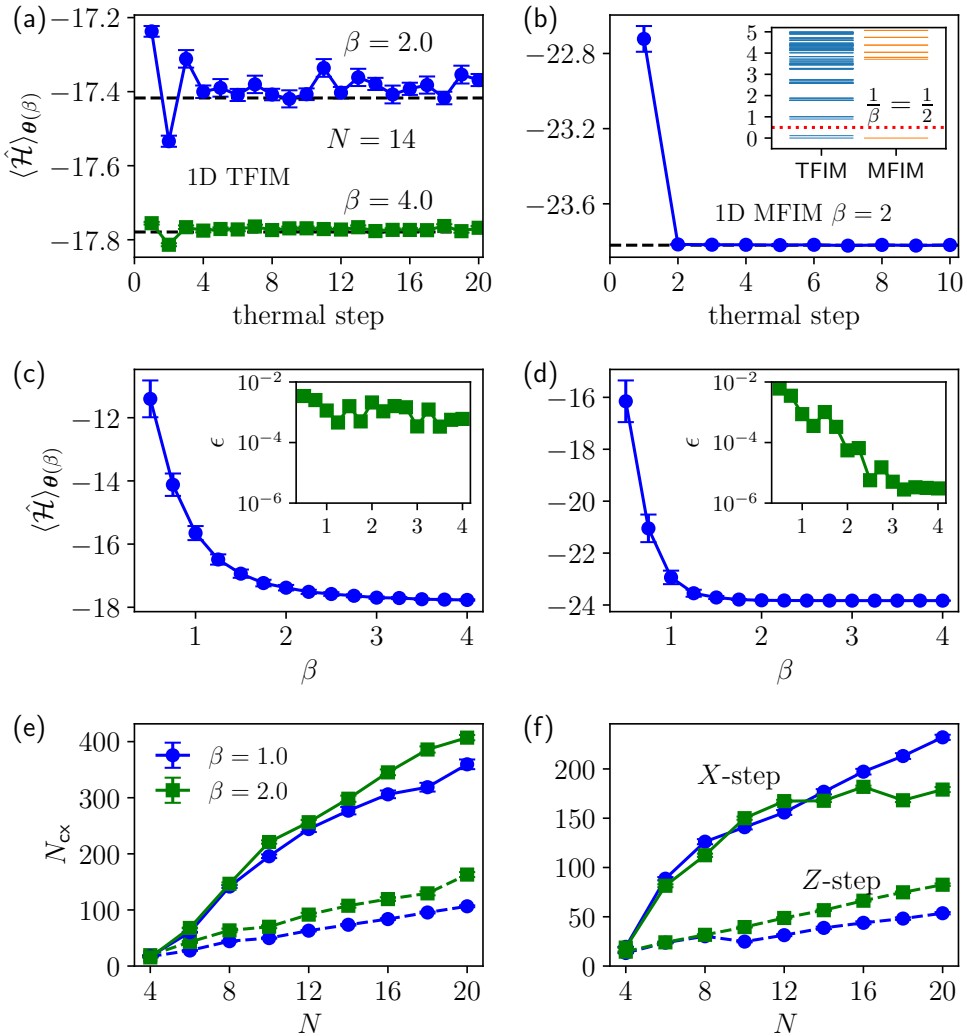

Figure 3: **AVQMETTS results for 1D TFIM and MFIM.** (a) AVQMETTS thermal energy $\langle\hat{\mathcal{H}}\rangle_{\boldsymbol{\theta}(\beta)}$ versus thermal step number for $N = 14$ TFIM. Different curves are for $\beta = 2$ (blue) and $\beta = 4$ (green). Each point is averaged over $S = 288$ samples, and the error bars denote the standard error. The estimator converges to the exact value (dashed) for larger step number. (b) Same quantity as in panel (a) for the $N = 14$ MFIM and $\beta = 2$. Inset shows energy level diagrams for the 14-site TFIM and MFIM. Red dashed line indicates approximate thermal energy at $\beta = 2$. (c,d) Thermal energies $\langle\hat{\mathcal{H}}\rangle_{\boldsymbol{\theta}(\beta)}$ as a function of inverse temperature $\beta$. The average and standard error are obtained for a METTS sample of size $S = 256$ for $\beta > 0.5$ and $S = 512$ otherwise. The first ten thermal steps are discarded. Insets show relative error $\epsilon$ of AVQMETTS compared to ED, which is below 0.01 throughout. (e,f) Average number of CNOT gates $N_{\text{cx}}$ in the AVQITE circuits associated with $X$ and $Z$-thermal steps as a function of system size $N$ that produce the states in the AVQMETTS sample at $\beta = 1, 2$. Panel (e) is for the TFIM and panel (f) is for the MFIM. Error bars denote standard deviation $\sigma_{\text{cx}}$. $N_{\text{cx}}$ scales approximately linearly with $N$, and we observe much larger values at $X$-thermal steps compared to at $Z$-steps in Fig. 2.

AVQMETTS sampling task much simpler than for the TFIM.

In Fig. 3(c) we plot the AVQMETTS estimated thermal energy $\langle\hat{\mathcal{H}}\rangle_{\boldsymbol{\theta}(\beta)}$ for the TFIM as a function of $\beta$ between $0.5 \le \beta \le 4$. We observe that the thermal energy decreases as a function of $\beta$ and converges to the ground state energy with increasing $\beta$. The inset shows the relative error compared to the exact thermal energy $\epsilon \equiv |1 - \langle\hat{\mathcal{H}}\rangle_{\boldsymbol{\theta}(\beta)}/\langle\hat{\mathcal{H}}\rangle_{\beta}|$, which lies between $10^{-2}$ and $10^{-4}$ for the values of $\beta$ we consider. A similar plot of $\langle\hat{\mathcal{H}}\rangle_{\boldsymbol{\theta}(\beta)}$ and its relative error $\epsilon$ is shown in Fig. 3(d) for the MFIM. The convergence to the ground state occurs faster due to the large gap of about $4J$ between the ground and the first excited states, and the error $\epsilon$ is also smaller. With a fixed ensemble size $S = S_{\mathrm{w}} = 256$ for $\beta > 0.5$ and $S = S_{\mathrm{w}} = 512$ otherwise in AVQMETTS calculations, the standard error of $\langle\hat{\mathcal{H}}\rangle_{\boldsymbol{\theta}(\beta)}$ generally grows with decreasing $\beta$ (or increasing $T$) as more states contribute to the thermal average.

To demonstrate the scalability of AVQMETTS calculations, we numerically study the required quantum resources for the algorithm as a function of system size $N$ at fixed temperatures. We use the number of CNOT gates $N_{\mathrm{cx}}$ in the ansatz circuits to quantify the required resources, which is the relevant figure of merit for NISQ hardware. Fig. 3(e) shows that $N_{\mathrm{cx}}$ grows approximately linearly with $N$ for the TFIM, with a slope that is increasing with $\beta$. A linear system-size scaling of $N_{\mathrm{cx}}$ is also observed for the MFIM, albeit with a smaller slope and magnitude compared with the TFIM. The error bars represent the standard deviation, which is defined as $\sigma_{\mathrm{cx}} = \sqrt{\frac{1}{S}\sum_{i=1}^{S}\left(N_{\mathrm{cx}}^i - \langle N_{\mathrm{cx}}\rangle\right)^2}$, where $\langle N_{\mathrm{cx}}\rangle$ is the average value. We find that the $X$-steps require a much larger number of CNOTs than the $Z$-steps. We note that in the $\beta \to \infty$ limit, the AVQMETTS calculation converges to AVQITE ground state calculation, where $N_{\mathrm{cx}}$ scales linearly and quadratically with $N$ for MFIM and TFIM at the critical point ($h_x/J = 1$), respectively [47]. Therefore, $N_{\mathrm{cx}}$ shows a better linear-$N$ scaling in AVQMETTS calculations of TFIM at criticality at nonzero temperature compared to at zero temperature. This improved $N$-scaling of $N_{\mathrm{cx}}$ is also observed in the real-time dynamics when comparing simulations at fixed final simulation time versus asymptotically long times [50]. Finally, in classical METTS calculations of 1D models the bond dimension of MPSs is expected to be independent of system size $N$ for gapped systems and of order $\mathcal{O}(N)$ at criticality [18]. Therefore, the classical computational complexity of METTS calculations for the 1D MFIM and TFIM is also expected to approach $\mathcal{O}(N)$ and $\mathcal{O}(N^2)$ as $\beta \to \infty$, respectively.

# 6  AVQMETTS results for 2D spin models

The classical METTS algorithm with DMRG as the imaginary time solver is efficient in calculating low-to-intermediate temperature properties of 1D systems, even though the sampling becomes costly for higher temperature [23]. Therefore, it is important to extend the AVQMETTS benchmark to 2D models, where DMRG suffers from an exponentially growing bond dimension even for area law states [18]. METTS+MPS simulations of 2D models have been progressed using cylindrical geometries with finite circumference, but the calculations are extremely demanding [58,59]. Even though tensor network-based simulations have made much progress in addressing this fundamental challenge [19,60], their application to METTS at finite temperature still remains difficult. The higher dimensionality $d > 1$ also introduces fundamentally different physics beyond 1D systems such as the stabilization of ordered phase at finite temperature and finite temperature phase transitions and criticality [52,53,61–63].

In this section, we apply AVQMETTS to thermal energy calculations for the 2D TFIM and MFIM on the different square lattice geometries up to $N = 4 \times 4$ with PBC in both directions [see Fig. 1(b)]. We consider the TFIM with $h_x = 3.05$ for which the ground state is doubly degenerate and exhibits a finite magnetization in the thermodynamic limit. At finite system sizes, however, there exists a splitting between the two lowest energy states, which becomes

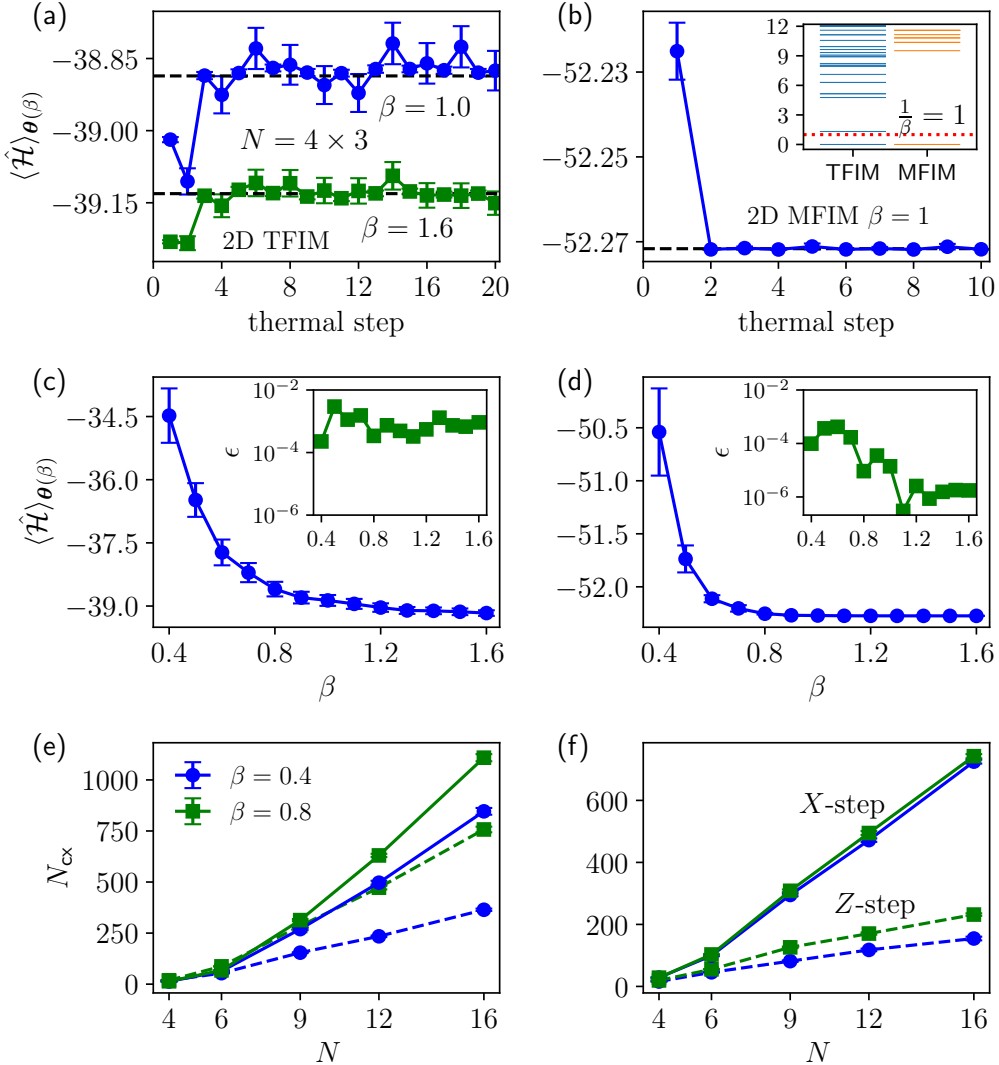

Figure 4: **AVQMETTS results for 2D TFIM and MFIM.** (a) Thermal energy $\langle\hat{\mathcal{H}}\rangle_{\boldsymbol{\theta}(\beta)}$ obtained from AVQMETTS versus thermal step in the $4\times 3$-site TFIM with $h_x = 3.05$ for two different inverse temperatures $\beta = 1$ and $\beta = 1.6$. The average is obtained from a METTS sample of size $S = 288$. Error bars denote standard error of the sample mean. (b) Same quantity as in panel (a) in the MFIM with $h_z = 1.525$ for $\beta = 1$. Inset shows low energy level diagram of the two models. (c, d) Thermal energies $\langle\hat{\mathcal{H}}\rangle_{\boldsymbol{\theta}(\beta)}$ versus $\beta$ for $4\times 3$-site TFIM (c) and MFIM (d). Standard error increases as $\beta$ decreases and more states contribute to the thermal average. Inset shows the relative error between AVQMETTS and ED. (e, f) Average number of CNOT gates $N_{cx}$ in the AVQMETTS circuits at $X$ and $Z$-thermal steps as a function of system size $N$ for $\beta = 0.4, 0.8$. Error bars denote standard deviation $\sigma_{cx}$. $N_{cx}$ at $X$-thermal steps are larger than that at $Z$-steps, consistent with 1D results.

exponentially small in the system size. For the $4\times 3$-site TFIM, the gap between the ground states and the first excited state is $\Delta E^{(\text{TFIM})} = E_{\text{ex},1} - E_{\text{GS}} \approx 1.3$. For the MFIM simulation we choose $h_x = 3.05$ and $h_z = 1.525$, which results in a rather large energy gap $\Delta E^{(\text{MFIM})} \approx 9.5$ at $N = 4 \times 3$. It is important to note that both models are nonintegrable in 2D and the two parameter sets are chosen to represent models with different gap sizes.

In Fig. 4(a, b), we present the convergence behavior of the thermal energy $\langle\hat{\mathcal{H}}\rangle_{\boldsymbol{\theta}(\beta)}$ with

increasing thermal step number for the TFIM in panel (a) at $\beta = 1.0$ and $\beta = 1.6$ and the MFIM in panel (b) at $\beta = 1.0$. The average energy $\langle \hat{\mathcal{H}} \rangle_{\theta(\beta)}$ is obtained for a METTS sample of size $S = 288$. The error bars denote the standard error of the mean, which is larger at smaller $\beta$ and also larger than for the 1D models. Similarly to the 1D model simulations, $\langle \hat{\mathcal{H}} \rangle_{\theta(\beta)}$ converges after the third thermal step for the TFIM, with residual fluctuations tied to the finite ensemble size $S = 288$ that reduce in amplitude with increasing $\beta$. Likewise, $\langle \hat{\mathcal{H}} \rangle_{\theta(\beta)}$ for the MFIM converges faster with smaller errors due to the presence of a much larger gap between the ground and first excited states, as illustrated in the inset of Fig. 1(b).

In Fig. 4(c, d), we present the AVQMETTS thermal energy estimation as a function of $\beta$ for the TFIM (in panel c) and for the MFIM (in panel d). Compared with the 1D model simulations shown in Sec. 5, we observe a similar convergence to the ground state with increasing $\beta$ as well as an increasing statistical error with decreasing $\beta$ (or increasing $T$). The relative error $\epsilon$ between $\langle \hat{\mathcal{H}} \rangle_{\theta(\beta)}$ and the ED results lies between $10^{-2}$ and $10^{-4}$ for the TFIM, and between $10^{-3}$ and $10^{-6}$ for the MFIM, as shown in the insets in Figs. 4(c) and (d). Since the energy gaps between the ground and the first excited states are much larger in the 2D models compared to the 1D models studied above, we here use a smaller $\beta$ range for the 2D models.

In Fig. 4(e, f) we plot the distribution of the number $N_{\mathrm{cx}}$ of CNOTs in the AVQMETTS circuits as a function of system size $N$ of different 2D square lattice geometries for the TFIM and MFIM at three temperatures $\beta = 0.4, 0.8$. We observe a general trend that $N_{\mathrm{cx}}$ grows with increasing $\beta$, which agrees with the results for 1D models shown in Fig. 3(e,f). In contrast, we observe an approximate linear to superlinear transition in the system size scaling of $N_{\mathrm{cx}}$ for the 2D TFIM. The range of system sizes is too small to make any definite statement of whether the scaling is polynomial or exponential. Interestingly, for the 2D MFIM, an approximate linear scaling remains, but the slope becomes larger than in 1D. This may be related to the large energy gap $\Delta E^{(\mathrm{MFIM})}$ for our choice of parameters in the MFIM for which the thermal average probes mostly ground state properties. We note that in the next Sec. 7, we focused on simulating the TFIM close to the thermal phase transition, where several excited states contribute to the partition function. Finally, the bimodal distribution of $N_{\mathrm{cx}}$ in the $Z$- and $X$-thermal steps is also observed in the AVQMETTS calculations of 2D models.

## 7  AVQMETTS estimation of critical temperature in 2D TFIM

The analysis performed in the previous sections establishes AVQMETTS as a viable method to study finite-temperature systems. Now we demonstrate one important application, which is the simulation of thermal phase transitions and the calculation of the associated transition temperature $T_c$.

We focus on the 2D TFIM, which exhibits a continuous phase transition at finite temperature $T$ and transverse field $h_x$. At zero field $h_x/J = 0$, this model becomes the classical 2D Ising model on the square lattice, which can be exactly solved analytically [53, 61]. In the thermodynamic limit, the system undergoes a phase transition from a low-temperature ferromagnetically (FM) ordered phase to a high-temperature paramagnetic (PM) phase at temperature $T_c/J = 2/\ln(1 + \sqrt{2})$. This transition is driven entirely by thermal fluctuations and defines the universality class of the $d = 2$ classical Ising model. A finite transverse field $h_x$ breaks the integrability of the 2D model. Stronger $h_x$ increases quantum fluctuations and thus reduces $T_c$. Using the Suzuki-Trotter method [64], the 2D model at finite temperatures can be mapped to an anisotropic classical model in three dimensions (with size dependent couplings) that can be investigated by Monte-Carlo methods [65–67]. The universality class of the transition at finite $T_c$ and $h_x$ is still that of the $d = 2$ classical Ising model. Once $T_c \to 0$ is suppressed to zero at the quantum critical point $h_x/J = 3.04438$, the transition is driven entirely by quan-

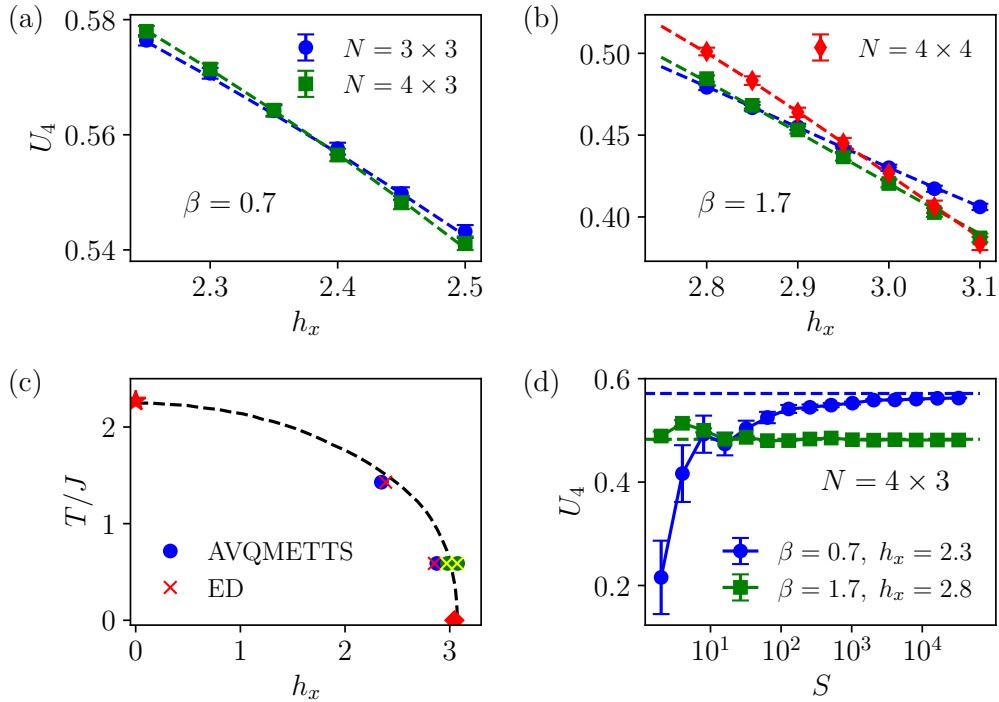

Figure 5: **Locating the thermal phase transition in 2D TFIM.** (a, b) The fourth-order Binder cumulant $U_4$ as a function of the control parameter $h_x$ for system sizes $N = 3 \times 3$ and $N = 4 \times 3$ at fixed inverse temperature values $\beta = 0.7$ (a) and 1.7 (b). Additional results for $N = 4 \times 4$ are also presented at $\beta = 1.7$ (b). The dashed lines correspond to the ED results, whereas the points are computed by AVQMETTS. The crossing of the two Binder cumulant curves determines the critical field $h_x^c$ at the corresponding temperature $T = 1/\beta$. The thus-obtained representative points $(h_x^c, T_c)$ on the critical line of the $h_x - T$ phase diagram of the 2D TFIM are plotted in panel (c). From left to right, the three points at $T_c = 1/1.7 = 0.59$ are estimates using data for $(3 \times 3, 4 \times 3)$, $(3 \times 3, 4 \times 4)$, and $(4 \times 3, 4 \times 4)$ sites. The points obtained from the latter two data sets are colored in green to distinguish them from the other blue representative points obtained from the $(3 \times 3, 4 \times 3)$ crossings. The dashed line is obtained from an approximate treatment of the exact series expansions [69–71]. The Onsager solution $(h_x = 0, T_c = 2/\ln(1 + \sqrt{2}) \approx 2.27)$, and the quantum critical point $(h_x = 3.04438, T_c = 0)$ are marked with a star and diamond, respectively. (d) Convergence of $U_4$ as a function of ensemble size $S$ relative to the ED values (dashed lines) for $(\beta, h_x^c) = (0.7, 2.3)$ (blue) and $(1.7, 2.8)$ (green).

tum fluctuations and the universality class of the transition is changed to that of the classical Ising model in $d = 3$ dimension [54, 68–74].

We use AVQMETTS to determine two points on the finite temperature FM-PM phase boundary in the $h_x - T$ plane. We calculate the fourth-order Binder cumulant [75, 76]:

$$U_4 = 1 - \frac{\langle m^4 \rangle}{3 \langle m^2 \rangle^2} \tag{9}$$

to accurately locate the phase transition, using the well-known fact that the Binder cumulant for different system sizes cross at $T_c$. Here $m = \frac{1}{2N} \sum_{i=1}^{N} Z_i$ is the average magnetization. Because of its scale invariance at the critical point, the Binder cumulant $U_4$ is one of the best ways to reduce finite-size effects in numerical simulations of phase transitions.

In Fig. 5(a, b) we show the Binder cumulant $U_4$ obtained using AVQMETTS simulations (symbols) as a function of transverse field $h_x$ for two different square lattice geometries of size $N = 3 \times 3$ (blue) and $N = 4 \times 3$ (green). The results in panel (a) are obtained at inverse temperature $\beta = 0.7$ and the one in panel (b) are obtained at $\beta = 1.7$. The average is over METTS samples of size $S(\beta = 0.7) = 6 \times 10^4$ and $S(\beta = 1.7) = 7 \times 10^3$ and the error bars denote the standard error of the mean. The AVQMETTS results show excellent agreement with ED calculations (dashed lines). The phase transition point can be estimated as the crossing between the two lines of different size, using the fact that $U_4(4 \times 3) > U_4(3 \times 3)$ in the ordered phase. It is obvious that a large number of samples is needed to reduce the standard error below the difference of the averages for the two system sizes. Using this procedure, we determine two $h_x^c$ values on the thermal phase transition at $T_c = 1/0.7$ and $1/1.7$, which are shown in Fig. 5(c). We see a close agreement between the AVQMETTS (blue circles) and ED (red cross) results for the $h_x^c$ values. To estimate the impact of finite-size effects on the determination of $h_x^c$ using the Binder cumulant approach, we add a data set for for $N = 4 \times 4$, as shown in Fig. 5(b). We choose $\beta = 1.7$ for the analysis because the required sample size for thermal averaging is much smaller than that at $\beta = 0.7$. The two additional estimates of $h_x^c$ determined by the crossings between the Binder cumulant curves of $N = 4 \times 4$, $N = 3 \times 3$ and $N = 3 \times 3$ are shown as green circles (AVQMETTS) and yellow crosses (ED) in Fig. 5(c). Specifically, we obtain $h_x^c \approx 2.97$ from the $N = 3 \times 3$ and $N = 4 \times 4$ data, and 3.07 from the $N = 4 \times 3$ and $N = 4 \times 4$ data, which are slightly larger than $h_x^c \approx 2.87$ based on $N = 3 \times 3$ and $N = 4 \times 3$ calculations. For reference, we also include a dashed curve for the complete critical line, that is taken from an approximate calculation using an exact series expansions [69–71]. The star symbol indicates the exact transition temperature at zero field, obtained from the Onsager solution.

The statistical error of an AVQMETTS ensemble average depends strongly on temperature, which we observed already in Secs. 5 and 6. In Fig. 5(d), we plot the METTS ensemble-averaged Binder cumulant $U_4$ as a function of ensemble size $S$ at the two transition temperatures $\beta = 0.7$ and $\beta = 1.7$. We find a consistent temperature-dependence with the lower temperature simulation at $\beta = 1.7$ converging already for $S \gtrsim 100$, while the higher temperature simulation at $\beta = 0.7$ requiring $S \gtrsim 10^4$ samples for convergence. The need for large ensemble sizes at higher temperatures limits the application of METTS to lower temperatures, which has also been noted in Ref. [25].

## 8  Noisy AVQMETTS simulations

In practical quantum computing, NISQ hardware is subject to various error sources besides the inherent sampling noise. These include coherent errors caused by imperfect gate operations, as well as stochastic errors due to qubit decoherence, dephasing, and relaxation. Here we investigate how these hardware imperfections affect AVQMETTS calculations using a noise model proposed by Kandala et al. in Ref. [77]. This model consists of an amplitude damping channel ($\Lambda_a[\rho] = \sum_{i=1}^{2} E_i^a \rho E_i^{a\dagger}$) and a dephasing channel ($\Lambda_d[\rho] = \sum_{i=1}^{2} E_i^d \rho E_i^{d\dagger}$), which act on the qubit density matrix after each single-qubit or two-qubit gate operation. The Kraus operators are defined as follows:

$$E_1^a = \begin{pmatrix} 1 & 0 \\ 0 & \sqrt{1-p^a} \end{pmatrix}, \qquad E_2^a = \begin{pmatrix} 0 & \sqrt{p^a} \\ 0 & 0 \end{pmatrix},$$
$$E_1^d = \begin{pmatrix} 1 & 0 \\ 0 & \sqrt{1-p^d} \end{pmatrix}, \qquad E_2^d = \begin{pmatrix} 0 & 0 \\ 0 & \sqrt{p^d} \end{pmatrix}. \tag{10}$$

The error rates $p^a = 1 - e^{-t_g/T_1}$ and $p^d = 1 - e^{-2t_g/T_\phi}$ depend on the gate time $t_g$, qubit relaxation time $T_1$, and dephasing time $T_\phi = 2T_1 T_2/(2T_1 - T_2)$, where $T_2$ is the qubit coherence

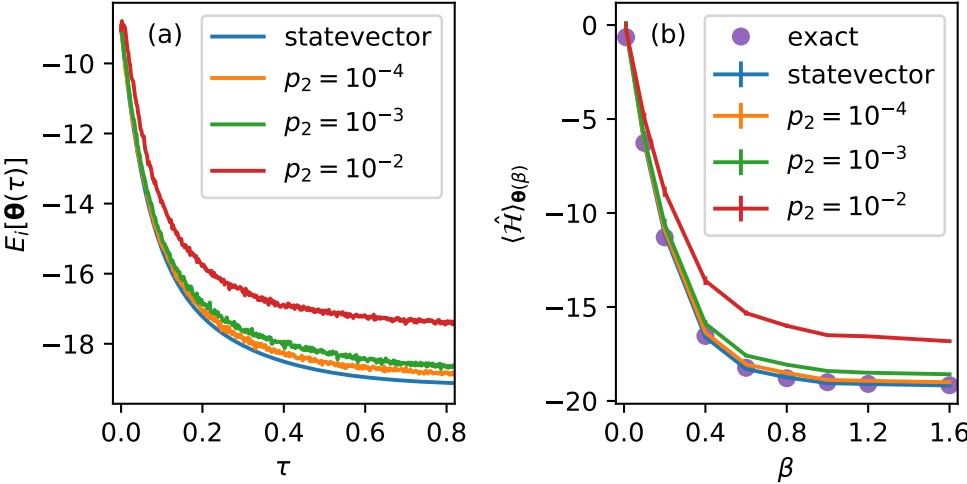

Figure 6: **Impact of noise on AVQMETTS simulations.** (a) Energy of an adaptive variational ansatz state, $E_i[\boldsymbol{\theta}(\tau)]$, along an imaginary time evolution path for the $N = 2 \times 3$ TFIM with $h_x = 3.05$. The simulation starts with a CPS in the $Z$-basis, $|i\rangle = |\downarrow\rangle^{\otimes N}$. We use $N_s = 2^{14}$ shots for each measurement circuit. To account for hardware noise effects, we adopt a noise model with a uniform single qubit error rate $p_1 = 10^{-4}$ and two-qubit gate error rate $p_2 = 10^{-2}$, $10^{-3}$, and $10^{-4}$. The noiseless statevector simulation result is also shown for reference. The variational ansatz totals $N_{\boldsymbol{\theta}} = 40$ parameters at the final imaginary time. (b) Thermal energy $\langle \hat{\mathcal{H}} \rangle_{\boldsymbol{\theta}(\beta)}$ obtained from AVQMETTS calculations with the same noise model as (a). The standard error of $\langle \hat{\mathcal{H}} \rangle_{\boldsymbol{\theta}(\beta)}$ is indicated by the vertical error bar, which becomes smaller than the line width at larger $\beta$. The METTS sample size is fixed to $S = 150$. The exact thermal energy is shown as circles for comparison. Note that, to estimate the thermal energy at inverse temperature $\beta$, the system is evolved from a CPS up to $\tau = \beta/2$ at each thermal step. A 7-qubit simulator with all-to-all connectivity, as in trapped-ion devices, is adopted for the calculations. Besides the 6 qubits used for modelling the physical lattice, an additional ancilla qubit is needed for the generalized Hadamard test to measure some elements of the quantum Fisher information matrix $M$ [47, 50].

time. Since $t_g$ depends on the gate being performed, this noise model assumes a different error rate for each gate. For simplicity in our analysis, we assume a uniform single-qubit gate error rate $p_1^a = p_1^d \equiv p_1 = 10^{-4}$, which closely matches the value observed in current hardware. To examine the impact of two-qubit gate noise, we also consider a uniform two-qubit error rate $p_2^a = p_2^d = p_2$, where $10^{-4} < p_2 < 10^{-2}$ [78, 79]. Additionally, we set $N_s = 2^{14}$ shots for each measurement circuit. We use a common set of CPSs and weights obtained from exact METTS calculations at each temperature. The crucial step of preparing METTSs from CPSs, however, is performed via noisy AVQITE simulations. Therefore, the results described below include the dominant noise effects and demonstrate the impact of noise on AVQMETTS calculations.

Figure 6(a) shows the variational energy, $E_i[\boldsymbol{\theta}(\tau)]$, as a function of imaginary time $\tau$ obtained from an AVQITE calculation, starting with an initial CPS $|i\rangle = |\downarrow\rangle^{\otimes N}$ in the $Z$-basis for an $N = 2 \times 3$ TFIM. If we set $\tau$ to end at $\beta/2$, this showcases a noisy simulation of one thermal step in AVQMETTS sampling at an inverse temperature $\beta$. The noisy simulation result at $p_2 = 10^{-4}$ closely follows that of the noiseless statevector simulator, with an energy deviation between the noisy and statevector simulation results $\Delta_E \approx 0.27(1.5\%)$ at $\tau = 0.4$ and $\Delta_E \approx 0.30(1.6\%)$ at $\tau = 0.8$. At a larger $p_2 = 10^{-3}$, the deviation increases modestly, with

$\Delta_{\mathrm{E}} \approx 0.57\,(3.1\%)$ at $\tau = 0.4$ and $\Delta_{\mathrm{E}} \approx 0.47\,(2.5\%)$ at $\tau = 0.8$. At $p_2 = 10^{-2}$, a typical value of the two-qubit gate error rate of current hardware, a more significant growth of the deviation is observed, with $\Delta_{\mathrm{E}} \approx 1.6\,(8.7\%)$ at $\tau = 0.4$ and $\Delta_{\mathrm{E}} \approx 1.7\,(8.8\%)$ at $\tau = 0.8$.

In Fig. 6(b) we plot the thermal energy $\langle \hat{\mathcal{H}} \rangle_{\boldsymbol{\theta}(\beta)}$ as a function of $\beta$ from the noisy AVQMETTS calculations. Consistent with the results for a single thermal step in Fig. 6(a), $\langle \hat{\mathcal{H}} \rangle_{\boldsymbol{\theta}(\beta)}$ agrees very well with the exact thermal energy $\langle \hat{\mathcal{H}} \rangle_{\beta}$ at $p_2 = 10^{-4}$. The thermal energy deviation, $\Delta_{\langle H \rangle} \equiv \langle \hat{\mathcal{H}} \rangle_{\boldsymbol{\theta}(\beta)} - \langle \hat{\mathcal{H}} \rangle_{\beta}$, is $0.28\,(1.5\%)$ at $\beta = 0.8$ and $0.16\,(0.8\%)$ at $\beta = 1.6$, compared with $\Delta_{\langle H \rangle} \approx 0.04\,(0.2\%)$ at $\beta = 0.8$ and $\Delta_{\langle H \rangle} \approx 0.01\,(0.07\%)$ at $\beta = 1.6$ for statevector simulations. When the noise level increases to $p_2 = 10^{-3}$, the deviation grows modestly, with $\Delta_{\langle H \rangle} \approx 0.72\,(3.8\%)$ at $\beta = 0.8$ and $\Delta_{\langle H \rangle} \approx 0.58\,(3.0\%)$ at $\beta = 1.6$. When the noise level further increases to $p_2 = 10^{-2}$, a substantial growth in deviation is noticeable. At $\beta = 0.8$, $\Delta_{\langle H \rangle}$ reaches about $2.8\,(14\%)$, while at $\beta = 1.6$, it reaches about $2.3\,(12\%)$.

## 9 Conclusion

In summary, we have developed an adaptive variational QMETTS approach (AVQMETTS) for finite-temperature quantum simulations that utilizes AVQITE for imaginary-time state propagation. This approach leverages the shallow and problem-specific quantum circuits that are generated by AVQITE that are suitable for simulations on NISQ hardware. We benchmark the performance of AVQMETTS for 1D and 2D TFIM and MFIM at different points in the phase diagram, including close to the quantum and thermal phase transitions.

For the 1D models, we report a high simulation fidelity with linear system-size scaling of the AVQMETTS circuit complexity, which we characterize by the number of CNOT gates required for preparing the METTSs. In the 2D TFIM we find a superlinear scaling of $N_{\mathrm{cx}}$ with $N$. Due to the limited range of system sizes that we can simulate classically, it remains an open question whether the scaling is exponential or polynomial. The latter case would provide a potential path towards quantum advantage, since DMRG exhibits an exponential scaling of the computational complexity due to an exponentially increasing bond dimension in 2D. For the 2D MFIM at a point in the phase diagram with a large energy gap between ground and first excited states, we find a linear scaling of $N_{\mathrm{cx}}$ with $N$. As expected, we observe that the number of required CNOT gates increases with dimension and decreasing temperature. Among the benchmark systems, the $4 \times 4$ TFIM at finite $T$ and at $h_x = 3.05$ requires the largest number of CNOT gates $N_{\mathrm{cx}} \approx 1200$ for AVQMETTS simulations of all the systems we have studied. Nevertheless, the computational load could be partially alleviated by the relatively small sample size $S$ needed in METTS at low temperatures, where a limited number of excited states contribute to the thermal expectation values. In contrast, at higher temperatures, AVQMETTS circuits become shallower, but one needs to use a significantly larger sample size $S$ for accurate thermal averaging. This is a known challenge of the METTS algorithm at larger temperatures.

We also apply the AVQMETTS algorithm to evaluate the transition temperature between the FM and PM phase at two points in the $h_x$-$T$ parameter plane of the 2D TFIM. We determine $T_c$ by accurately computing the fourth-order Binder cumulant $U_4$, which requires a METTS sample of size $S \approx 10^4$ at $\beta = 0.7$. This computational demand can be partially addressed by parallelizing the sampling in terms of independent Markov random walks used in the stochastic sampling algorithm. Our results are in excellent agreement with ED results inferred from the same finite systems.

Finally, we perform noisy AVQMETTS simulations for an $N = 2 \times 3$ TFIM to investigate the impact of noise on the results. At the noise level of $p_2 = 10^{-2}$ representative for the current hardware, the relative thermal energy error is about 12%. The relative error is reduced to about 3% when $p_2$ improves to $10^{-3}$.

The next steps are to perform AVQMETTS calculations on quantum hardware, which will require leveraging ongoing efforts in circuit optimizations for the AVQITE algorithm as well as making use of error mitigation techniques [80–85]. The distribution of quantum measurement shots among the various circuits used to obtain the AVQITE equations of motion (5) can also be optimized to reduce the total shot budgets [86]. Recently, practical error mitigation techniques such as zero-noise extrapolation have been demonstrated on quantum hardware to scale to circuits containing up to 26 qubits and 1080 CNOTs [87, 88]. Provided further improvements of the quantum hardware, we envision that AVQMETTS will enable calculations of thermodynamic properties for a wide range of quantum many-body systems. We consider this approach to be particularly pertinent in two dimensions, where the rapid growth in bond dimension with increasing system size or decreasing temperature poses a fundamental challenge to classical simulations. In addition to static observables at finite temperature, the formalism can also be naturally extended to the simulation of finite-temperature dynamical correlation functions [38]. It can therefore also be used as an impurity solver for finite-temperature quantum embedding simulations of real materials [15, 79, 89–91].

## Acknowledgements

This work was supported by the U.S. Department of Energy (DOE), Office of Science, Basic Energy Sciences, Materials Science and Engineering Division, including the grant of computer time at the National Energy Research Scientific Computing Center (NERSC) in Berkeley, California. The research was performed at the Ames National Laboratory, which is operated for the U.S. DOE by Iowa State University under Contract No. DE-AC02-07CH11358.

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
