# Peer review of "Adaptive variational quantum minimally entangled typical thermal states for finite temperature simulations"

_SciPost Physics, doi:SciPost Phys. 15, 102 (2023)_

## Round 1 · Referee Report · Anonymous · 2023-2-17

Report
Thanks to the authors for this interesting work. Indeed, implementing imaginary-time evolution on the quantum circuit has been a challenging task for the community. In this work, the authors propose the AVQITE algorithm, which can help to measure the thermal expectation value of an observable and indicate that this algorithm can be implemented on noisy quantum simulators. However, I worry that it can be difficult to implement the authors’ method on the present quantum computer. In this case, I do not recommend the publishment of this work in SciPost Physics, unless the authors show that it is promising to carry their results on the real noisy quantum simulator. My reasons are listed as follows
(a) For any model on a noisy quantum computer, a critical factor of robust simulations is the depth of a quantum circuit. Generally, we focus on the number of CX gates. Such things are shown in FIG3 and FIG4 of this work. Here, an important point is that the number of CX gates required by the authors’ method is about hundreds, which is not tolerable for the present noisy quantum simulator, e.g., IBM Quantum device. In this case, the authors need to show how to use optimization methods to compress their circuit to a safe depth.
(b) For the models shown in FIG1, the authors consider periodic boundary conditions (PBCs). Here, I suppose that the authors need to take the geometry of real noisy quantum simulators into account and show how to realize a model under PBCs based on a next-nearest device.
(c) For the operation for imaginary-time evolution U_{AVQITE}, the authors consider the decomposition based on all-to-all qubit connectivity. I do not think any present noisy quantum simulator supports this kind of geometry. If the authors would like to convince that their method is suitable for the present quantum device, they need to do the decomposition of their pseudo-Trotter step as Eq. 4 in terms of the special geometry of today’s quantum device.
In sum, There is a gap between the noiseless and noisy simulation. To faithfully convince the community, noisy simulations of a minimal case might be needed. According to the above reasons, I cannot recommend this manuscript for publication and think the authors need to address a critical question why their algorithm is suitable for the current-day quantum device.

---

## Round 1 · Referee Report · Anonymous · 2023-3-14

Strengths
1- offers o new original approach to simulate the imaginary time evolution of quantum many-body systems on quantum hardware.
2- it can be applied to systems with sign problem and in 2D
Weaknesses
1- The relative errors are not necessarily small and the method may not represent a killer app; i.e. it's not state of the art. (still, nice progress in the right direction).
Report
The authors discuss a generalization of the previously developed AVQITE time evolution algorithm to the case of minimally entangled typical thermal states. They test their newly minted AVQMETTS method through numerical simulations in (small) 1D and 2D systems and they offer a number of benchmarks to characterize its performance. The idea is original and holds potential for applications in NISQ devices. I therefore recommend it for publication after some minor concerns are addressed:
1- The authors refer to previous work for details on the algorithm and only briefly describe it without much depth. This would be fair if it wasn't because the method is quite new, leaving room for confusion. For instance, a naive question would be: Does the quantum circuit have to be optimized for each initial random state? I assume that's the case but, otherwise, if the circuit is obtained before hand, how is it done? Can the authors please clarify the necessary points to make this more clear?
2-It is not obvious why the algorithm should use initial product states (hence, METTS). Wouldn't it equally work with any initial random state?
See, for instance, the approach referred-to as ``canonical thermal pure quantum states''(CTPQS) (doi="10.1007/978-981-10-1506-9_3") and references therein.
Or is it that one uses initial product states to facilitate the implementation in a quantum device?
On the other hand: if one starts from an (entangled) initial random state, how will it affect the depth of the circuit and the performance of the method?
Requested changes
See point 1 in the report above.
Fidelity (nor infidelity) are never defined in the text, and are only defined in the caption of Fig. 2. It would better serve the reader to have it defined in the corresponding place in the text.

---

## Round 2 · Referee Report · Anonymous (Referee 1) · 2023-5-30

Report

In this revision, the authors have worked on my comments carefully. In particular, they consider a realistic noise model and investigate the noise on simulations. The paper has significantly improved. Thus, I now recommend it for publication.

---

## Round 2 · Referee Report · Anonymous (Referee 2) · 2023-6-21

Report

The authors have significantly improved the manuscript and addressed all the issues raised in my original report. I therefore recommend it for publication as is.

---

## Round 2 · Author Response

Dear Editor,

Many thanks for returning the reviews of our paper entitled "Adaptive variational quantum minimally entangled typical thermal states for finite temperature simulations.'' The reviewer in the invited report recommends our manuscript for publication with minor concerns to be addressed. The reviewer in the contributed report finds our work interesting, and suggests noisy simulations of a minimal model to be added in the manuscript.

To address the questions we have revised our manuscript to include
(a) A new section (VIII) for AVQMETTS simulations with noise model.
(b) Discussions to clarify some details of AVQITE, typical pure state-based approaches, among others.

We are confident that these substantial improvements address all the comments and questions. We therefore would like to seek further consideration of our work in SciPost Physics.

We look forward to hearing from you soon.

Yours sincerely,
Yongxin Yao on behalf of all the authors

Notice: This work was produced by Iowa State University under Contract No. DE-AC02CH11358 with the U.S. Department of Energy. Publisher acknowledges the U.S. Government license and the provision to provide public access under the DOE Public Access Plan (http://energy.gov/downloads/doe-public-access-plan).

---

## Round 2 · List of Changes

Response to the Invited Report

Referee's comment: The authors refer to previous work for details on the algorithm and only briefly describe it without much depth. This would be fair if it wasn't because the method is quite new, leaving room for confusion. For instance, a naive question would be: Does the quantum circuit have to be optimized for each initial random state? I assume that's the case but, otherwise, if the circuit is obtained before hand, how is it done? Can the authors please clarify the necessary points to make this more clear?

Reply: It is correct that each different initial random state, specifically the classical product state in this study, requires a unique AVQITE calculation. However, reusing AVQITE circuits is also feasible and helpful to reduce the quantum resource demand for AVQMETTS because of the duplicated CPSs during calculations.

To spell out the point clearly, we inserted the following sentence during the first exposition of AVQITE in Sec. II: ``\textcolor{blue}{We note that the resulting AVQITE circuit is problem-specific and tied to the initial state $\ket{i}$.}'' We also added the following lines in the description of AVQMETTS flowchart:``\textcolor{blue}{As the AVQITE circuit is associated with the initial state, each distinct CPS requires a unique AVQITE calculation. However, reusing AVQITE circuits is also feasible and can help minimize the quantum resource demand for AVQMETTS calculations. This is due to the fact that a CPS obtained from state collapse after a thermal step may be identical to one that was sampled in a previous step due to the inherent structure of the distribution of METTS. In the numerical simulations reported below, we observe $10-60\%$ of CPSs are sampled for more then once, depending on the system size and temperature.}''

Referee's comment: It is not obvious why the algorithm should use initial product states (hence, METTS). Wouldn't it equally work with any initial random state? See, for instance, the approach referred-to as canonical thermal pure quantum states''(CTPQS) (doi=``10.1007/978-981-10-1506-9\_3'') and references therein. Or is it that one uses initial product states to facilitate the implementation in a quantum device? On the other hand: if one starts from an (entangled) initial random state, how will it affect the depth of the circuit and the performance of the method?

Reply: Indeed, we use initial product states to facilitate the implementation in a quantum device. Furthermore, the circuit complexity for METTS preparation is bounded by that of the ground state, which has area-law entanglement (up to logarithmic violations for gapless systems), in contrast to the volume-law entanglement of a generic random state. The preparation of a truly random state is exponentially hard on quantum computers, hence quantum algorithms utilizing such states are not scalable.

We added the following lines in the Introduction section to clarify the point: ``\textcolor{blue}{We note that, while it might seem straightforward and appealing to leverage the statistical approach based on TPQ states in a quantum algorithm, the necessary initial step of random state preparation is known to be exponentially hard on quantum computers~[40].}''

Response to the Contributed Report

Referee's comment: For any model on a noisy quantum computer, a critical factor of robust simulations is the depth of a quantum circuit. Generally, we focus on the number of CX gates. Such things are shown in FIG3 and FIG4 of this work. Here, an important point is that the number of CX gates required by the authors’ method is about hundreds, which is not tolerable for the present noisy quantum simulator, e.g., IBM Quantum device. In this case, the authors need to show how to use optimization methods to compress their circuit to a safe depth.

Reply: The number of CNOT gates for circuit simulations of quantum systems will generally grow with system size. Our work demonstrates a favorable approximately linear system-size scaling for the AVQMETTS calculations of the studied spin models. The AVQITE algorithm we use to prepare METTS has been demonstrated to generate highly compact problem-specific ground state ans\"atze comparable to other state-of-the-art approach such as qubit-ADAPT-VQE. We agree that executing circuits with hundreds of CNOTs is challenging on present hardware, but not impossible given the rapid development of quantum computing technology. For instance, a recent publication (Y. Kim, et al, Nature Physics 19, 752–759 (2023) from the IBM group has demonstrated hardware calculations with circuits containing up to 1080 CNOTs.

We added the following line in the concluding remarks: ``\textcolor{blue}{Recently, practical error mitigation techniques such as zero-noise extrapolation have been demonstrated on quantum hardware to scale to circuits containing up to $26$ qubits and $1080$ CNOTs~[85,86].}''

Referee's comment: For the models shown in FIG1, the authors consider periodic boundary conditions (PBCs). Here, I suppose that the authors need to take the geometry of real noisy quantum simulators into account and show how to realize a model under PBCs based on a next-nearest device.

Reply: For digital quantum computers with restricted qubit connectivity, such as the superconducting transmon qubit-based processors, any nonlocal gate can still be implemented with the aid of a sequence of swap gates, because single-qubit gates plus the CNOT gates between nearest neighbors constitute a universal complete gate set. Therefore, except for some overhead of swap gates, the models with PBC can be simulated on digital quantum hardware. Moreover, in some circumstances the swap overhead is unnecessary. For example, for 1D models many devices (such as IBM QPUs with heavy-hexagon connectivity) have a sublattice matching the connectivity requirements of PBC. More generally, some quantum computing platforms (e.g. trapped ions, see below) have all-to-all connectivity.

Referee's comment: For the operation for imaginary-time evolution $U_\text{AVQITE}$, the authors consider the decomposition based on all-to-all qubit connectivity. I do not think any present noisy quantum simulator supports this kind of geometry. If the authors would like to convince that their method is suitable for the present quantum device, they need to do the decomposition of their pseudo-Trotter step as Eq. 4 in terms of the special geometry of today’s quantum device.

Reply: Trapped-ion quantum computers, such as Quantinuum’s System Model H1 and H2, allow to reorder and reconfigure the chains of ions within the architecture, enabling all-to-all connectivity. For the sake of simplicity, our CNOT-gate analysis is based on hardware with all-to-all connectivity. Some difference is expected when the analysis is switched to specific superconducting qubit devices; however, similar leading behaviour of system-size scaling should still hold. We updated the following line in the caption of Fig.2: ``\textcolor{blue}{$N_\text{cx}$ is calculated assuming all-to-all qubit connectivity as in trapped-ion quantum processors...}''

Referee's comment: In sum, There is a gap between the noiseless and noisy simulation. To faithfully convince the community, noisy simulations of a minimal case might be needed. According to the above reasons, I cannot recommend this manuscript for publication and think the authors need to address a critical question why their algorithm is suitable for
the current-day quantum device.

Reply: To address the referee's comment, we added a new section VIII on noisy AVQMETTS simulations of a $N=2\times 3$ TFIM. In the noisy simulation, we consider a realistic noise model and investigate the impact of noise on the thermal energies. We refer the referee to the main text for the details.

---

## Editorial Decision

published